# Effects of a Recombinant Gonadotropin-Releasing Hormone Vaccine on Reproductive Function in Adult Male ICR Mice

**DOI:** 10.3390/vaccines9080808

**Published:** 2021-07-21

**Authors:** Ai-Mei Chang, Chen-Chih Chen, Ding-Liang Hou, Guan-Ming Ke, Jai-Wei Lee

**Affiliations:** 1International Program in Animal Vaccine Technology, International College, National Pingtung University of Science and Technology, Pingtung 91201, Taiwan; j10685002@g4e.npust.edu.tw (A.-M.C.); kegm@mail.npust.edu.tw (G.-M.K.); 2Research Center for Animal Biologics, National Pingtung University of Science and Technology, Pingtung 91201, Taiwan; 3Institute of Wildlife Conservation, College of Veterinary Medicine, National Pingtung University of Science and Technology, Pingtung 91201, Taiwan; 4Department of Veterinary Medicine, College of Veterinary Medicine, National Pingtung University of Science and Technology, Pingtung 91201, Taiwan; belicekevin@gmail.com; 5Institute of Animal Vaccine, College of Veterinary Medicine, National Pingtung University of Science and Technology, Pingtung 91201, Taiwan; 6Department of Tropical Agriculture and International Cooperation, International College, National Pingtung University of Science and Technology, Pingtung 91201, Taiwan

**Keywords:** gonadotropin-releasing hormone, immunocontraception, fertility control, mammal

## Abstract

Gonadotropin-releasing hormone (GnRH) regulates the reproductive endocrine system in mammals. The GnRH immunocontraception vaccine can aid animal population control and management. We evaluated a recombinant GnRH fusion protein with the adjuvant MONTANIDE ISA 206 VG as a GnRH vaccine in adult male ICR mice by evaluating anti-GnRH antibodies; concentrations of follicle-stimulating hormone (FSH), luteinizing hormone (LH), and testosterone; testis size and histomorphology; and semen quality. Response was assessed after intramuscular administration of the vaccine to mice in weeks 0, 4, and 8. The vaccine induced specific antibody response by week 5, with peak of antibody levels observed by week 13 and a declining level thereafter until the end of the study at week 24. Furthermore, it reduced serum FSH, LH, and testosterone concentrations. The vaccinated mice exhibited testicular atrophy and reduced sperm quality, concentration, morphology, and viability compared to control males. The outcomes of pairings of treated males with untreated females revealed reduced mating, pregnancy rates and number of litters compared to control pairings. Assessment of this GnRH vaccine in different species could assist its development for future applications.

## 1. Introduction

The overpopulation of free-living domestic or wild animals has become a serious social issue worldwide [1,2,3]. For example, the estimated global dog population is approximately 700 million, of which 300 million are street dogs [4]. The overpopulation of free-living dogs threatens human society because they can transmit pathogens, attack livestock or humans, cause traffic accidents, pollute the environment, and exhibit aggravating behavior, such as barking. Overpopulation of free-living wild animals was reported in a certain area in many countries, for instance, feral horses and kangaroos in Australia [5,6], white-tail deer in the USA [7], Formosan macaques in Taiwan [8] and some large predators in southern Africa [3]. The overpopulation of free-living wild animals may cause widespread environmental degradation, eliminate populations of native species, damage to crops, or even conflict with humans [8,9,10]. To mitigate the conflicts between humans and other free-living animals, population control management and strategies are needed for specific animals [11].

Culling and other non-lethal methods, such as translocation and fertility control would be options for management [12,13,14]. Culling, the traditional method of animal population control, is usually ineffective as it has only a short-term impact on population densities [15]. According to the research of Pepin et al. [16] on the population of wild pig, culling alone was not being able to reduce the populations substantially. In grey squirrel and red fox population control programs, the recovered numbers of population and compensation process via immigration from surrounding areas were noticed after culling [17,18]. In addition, translocation of animals might increase the risk of disease, change survival rates, and behaviors [19]. Current approaches for animal population management are majorly focusing on fertility control, such as surgical sterilization are generally accepted but relatively expensive, time consuming, and frequently criticized by animal welfare organizations [20]. Presumably, it is more effective to combine fertility control and culling methods to rapidly decrease number for a population [21]. Accordingly, more humane alternatives, such as immunocontraception or immunocastration, have been attracting interest [22].

Unlike vaccines against infectious pathogens, most immunocontraception vaccines are designed to induce immune responses against self-antigens, among which gonadotropin-releasing hormone (GnRH) is a major target for development immunocontraception vaccines. GnRH is a key hormone secreted by the hypothalamus that regulates the reproductive system in mammals [23]. Blocking GnRH inhibits the downstream release of luteinizing hormone (LH) and follicle-stimulating hormone (FSH) from the pituitary gland, which in turn impedes ovulation in females and spermatogenesis in males. Therefore, a GnRH-based contraception vaccine can be used in both males and females [24,25]. Thus far, the structure of GnRH has been identified in 23 forms in vertebrates [26]. The amino acid sequence of GnRH is highly conserved among phylogenetically closed species [27]. For example, mammalian GnRH (or GnRH-I), except for guinea pigs, has identical amino acid sequences (pGlu-His-Trp-Ser-Tyr-Gly-Leu-Arg-Pro-Gly-NH_2_) [26].

Several commercial GnRH vaccine formulations have been developed, such as GonaCon (National Wildlife Research Center, Fort Collins, CO, USA) and Improvac (Zoetis South Africa, Sandton, South Africa). GonaCon comprises GnRH conjugated with a carrier protein as the antigen and purified *Mycobacterium avium* as the adjuvant. It has been approved for use in white-tailed deer by the US Environmental Protection Agency (EPA) [28,29,30]. Improvac, prepared with gonadotropin-releasing factor linked to a carrier protein, is approved by the US Food and Drug Administration for use in the prevention of boar taint [31]. Several products for fertility control are currently underdeveloped and may be commercially available the near future [13]. However, the side effects elicited by immunization observed in different species must be seriously considered. Vargas-Pino et al. [32] reported that administration of GonaCon in dogs induced hyperthermia, as well as swelling and muscular atrophy at the site of injection. Thus, evaluating adjuvant–antigen combinations for a specific species is necessary for minimizing or even preventing side effects.

GnRH-based vaccines inhibit fertility in both male and female mammals [33,34] However, the persistency of antibody responses varies, presumably due to the immunogenicity of the modified GnRH–protein conjugation. Herein, we adopted a recombinant antigen—containing eight repeated GnRH motifs, fused with four T-helper epitopes, and formulated with commercial adjuvant Montanide ISA206 (water-in-oil–water emulsions, W/O/W)—which was emulsified with varied antigens, and no obvious side effects were noticed. We think this adjuvant could be useful for canine and other species. The efficacy of this GnRH-based vaccine was investigated in sexually matured male mice (30-week-old) in the present study. Parameters analyzed included serum anti-GnRH antibody levels; concentrations of sex hormones (FSH, LH, and testosterone); testis size, and histomorphology; semen quality; and outcomes of pairings with untreated females.

## 2. Materials and Methods

### 2.1. Animals

Twenty 30-week-old male adult Institute of Cancer Research (ICR) mice were obtained from BioLASCO (Taipei, Taiwan). The 30-week-old ICR mice were considered as middle-aged (the average lifespan is 73.8 weeks) [35,36] which were the target age for our study. The mice were housed in a certified laboratory animal center at the National Pingtung University of Science and Technology in Taiwan. Food and water were provided ad libitum. The experimental procedures were approved by the Institutional Animal Care and Use Committee of the National Pingtung University of Science and Technology (approval number: NPUST-107-017).

### 2.2. Vaccine Preparation

The antigen of GnRH-based vaccine contained a multimer structure fusion protein that coded eight repeats of GnRH-I and four T-helper epitopes. The genetic fragments and complete amino acid and DNA sequences are illustrated in Figure 1. The sequence was synthesized by Taiwan Biomedical (Taipei, Taiwan). T-cell epitopes were designed based on the *Plasmodium falciparum* (T1), Tetanus toxoid (T2), respiratory syncytial virus (T3), and measles virus (T4), respectively. The conjugation of the fusion protein was modified from that used by Talwar et al. [37], with 8 repetitions of the GnRH sequence and the recombinant antigen being expressed by the *Escherichia coli* expression system. Finally, the stock GnRH antigen (2 mg/mL) emulsified with an equal volume of oil-based MONTANIDE ISA 206 VG adjuvant (final concentration of GnRH antigen was 1 mg/mL).

### 2.3. Immunization

The mice were randomly assigned to the control group or the vaccinated group. Mice in the vaccinated group (*n* = 10) were intramuscularly injected (0.1 mL, 100 μg) in both left and right hindquarters with the GnRH vaccine (total dosage was 0.2 mL, 200 ug GnRH antigen for immunization). A vaccine booster was administered 4 and 8 weeks after the primary immunization. However, mice in the control group (*n* = 10) received intramuscular injections of 0.2 mL sterile phosphate-buffered saline (PBS) with adjuvant on the same schedule of immunization as that followed by the vaccinated group.

### 2.4. Sample Collection

Peripheral blood samples (100–200 μL) were collected from the submandibular or facial vein of each mouse every 2 weeks. The blood samples were allowed to clot in a nuclease-free tube for at least 30 min and centrifuged (1000× *g*) for 10 min. The serum was carefully collected and stored at −20 °C until analysis.

Twenty-four weeks after the primary immunization, six mice were randomly selected from each group. The mice were anesthetized with 4% isoflurane and killed by cervical dislocation. The heart was punctured to collect 1 mL of blood into a nuclease-free microcentrifuge tube. Next, the testis was dissected, weighed, and placed in Bouin’s solution for 24 h for tissue fixation. Sperm was collected from the cauda epididymis and stored in CARD mHTF, modified human tubal fluid (Cosmo Bio, Japan) [38].

### 2.5. ELISA Measurement of Anti-GnRH IgG Antibody

Flat-bottom 96-well Nunc 469,949 Immuno Clear Standard Module plates (Intermed, Nunc, Gibco, Burlington, Ontario, Canada) were coated with 1 μg of GnRH protein in coating buffer (100 μL/well, 0.1 M carbonate bicarbonate buffer, pH 9.6; Sigma Aldrich, Australia) and incubated overnight at 4 °C. Next, the wells were blocked by BlockPRO Buffer (200 μL/well; Visual Protein, Taiwan) at 37 °C for 1 h. The blocking buffer was discarded, and diluted serum (50 μL/well; 1:50) samples were added in duplicate. After washing five times with buffer, specifically PBS with 0.05% (*v*/*v*) Tween 20, the plates were incubated at 37 °C for 1 h. This was followed by the addition of HRP-conjugated goat anti-mouse IgG antibody (1:5000; Abcam, UK) in PBS containing 0.5% bovine serum albumin (50 μL/well). The plates were washed five times, and 100 μL of 3,3′,5,5′-tetramethylbenzidine dihydrochloride (TMB) substrate (Sigma Aldrich, Australia) was added to each well and allowed to react for 10 min. The reaction was stopped through the addition of 100 μL of 2 M sulfuric acid solution to each well. Finally, the absorbance of each well was measured at 450 nm with a Multiskan FC Microplate Photometer (Thermo Scientific, Waltham, MA, USA). The results are presented as optical density (OD) values considered positive if they exceeded the mean antibody level plus three times the standard deviation (SD) at the reading in week 0 [39].

### 2.6. Measurement of Serum LH/FSH Concentrations

Serum samples from day 0 and weeks 4, 16, and 24 were used to measure LH and FSH concentrations. The LH and FSH hormone levels were measured on a Luminex 200IS platform using a MILLIPLEX MAP Mouse Pituitary Magnetic Bead Panel (Millipore, Germany) [40], according to the manufacturer’s instructions.

### 2.7. Measurement of Serum Testosterone Concentrations

To avoid interference from other steroid compounds, the serum was processed before testosterone concentration determination. In brief, 500 μL of serum sample was mixed thoroughly with 2500 μL of diethyl ether and allowed to stand to enable mixture separated into two layers. Subsequently, the upper layer was carefully transferred into a clean test tube with a Pasteur pipette. The procedure was repeated three times. Finally, the diethyl ether was evaporated by heating to 30 °C under a gentle stream of nitrogen, and the residue extract was stored at −20 °C until use. The serum sample was measured in week 24 using a testosterone enzyme-linked immunosorbent assay (ELISA) kit (Cayman Chemical, Ann Harbor, MI, USA), according to the manufacturer’s instructions. The measurements of all samples were performed in duplicate, and each duplicate was determined twice with the same ELISA kit.

### 2.8. Intra-Assay and Inter-Assay Variation

To show the precision and repeatability of our experiments. The intra-assay and inter-assay variations in the GnRH IgG antibody and the serum LH, FSH, and testosterone concentrations were measured for quality control in each analytical batch, expressed as the coefficient of variation (CV).

The intra-assay described the variation of data from one experiment. The CV calculation in intra-assay was the average value from each sample in duplicated. The inter-assay was a plate-to-plate consistency. The CV calculation of mean values for the low and high concentration standards on each plate.

Mean, standard deviation (SD), and CV (SD/mean × 100%) were calculated. An assay was defined as valid when the intra-assay and inter-assay CV was less than 10% and 15%, respectively [41].

### 2.9. Analysis of Testis Tissue and Sperm Quality

#### 2.9.1. Testis Morphology Measurement

After euthanasia, the testes were dissected for morphological measurement. The vessel, adipose, and connective tissues were removed. The body and testis weights were recorded, and the external sizes, length, and width were measured using a digital caliper. The testis volume was calculated based on the formula for a prolate spheroid volume (mm^3^) = 4/3 π [0.5 × testis width/2]^2^ [testis length/2]) [42].

#### 2.9.2. Histological Examination of the Testis

The testis tissue was fixed in Bouin’s solution for 24 h. After fixation, the tissue was washed and then passed through graded alcohols. Finally, the tissue was embedded in a single paraffin wax block for histological evaluation. Sections (5 μm) were mounted onto a glass slide and dried by heating at 37 °C for 24–36 h. The slide section was deparaffinized, rehydrated through a graded series of ethanol baths, and stained using Gill’s hematoxylin and eosin stain. The slides were left to dry in an incubator at 37 °C for 4–8 h and mounted with a coverslip. The slides were examined under a microscope at 10× and 40× magnification for morphological observation.

#### 2.9.3. Sperm Collection

The epididymis of the euthanized mice was removed for semen collection. The caudal part of the epididymis was carefully trimmed to remove adipose and connective tissues, rinsed in PBS, and placed in 200 μL of CARD mHTF medium. Each cauda was cut into 4–6 sections, and semen was released into the medium by incubation for 1 h at 37 °C under 5% CO_2_ [38]. After incubation, the tissue was removed, and the suspension was mixed gently by pipetting. The suspension was used for morphological observation and sperm quality analysis.

#### 2.9.4. Analysis of Sperm Quality

The sperm quality was analyzed based on sperm concentration, abnormality, and viability. Sperm count was measured using a Neubauer hemocytometer [43], and sperm viability was assessed through wet preparation microscopy (20 μm). Sperm morphology was evaluated using air-dried Giemsa staining [44] and eosin–nigrosin staining, a wet staining method [45], to determine the proportions of normal and abnormal sperm. For Giemsa staining, a 10-μL aliquot of semen diluted in PBS was smeared over a microscopic slide. After air drying, the smear was fixed with methanol for 5 min, stained with Giemsa for 15 min, and finally washed with tap water to remove the debris [44]. For eosin–nigrosin staining, a 5-μL aliquot of semen diluted in PBS was mixed with an equal volume of eosin–nigrosin solution [45]. The mixture was incubated at room temperature for 1 min and smeared onto a new slide. Next, 200–500 sperms from the semen sample of each mouse were assessed for morphological abnormality—specifically, sperm head evaluation according to the criteria described by Wyrobek et al. [46]. Sperm without a tail or a head that was in contact or overlaid with other sperm or debris were excluded from the evaluation.

### 2.10. Fertility Test

A mating test was conducted to evaluate fertility after immunization. Vaccinated (n = 4) and control group (n = 4) males were housed with a sexually mature female mouse (age 54 weeks) in week 24. The female mice were examined for the presence of vaginal plugs every morning as evidence of mating. If a plug was observed, the female mouse was housed individually for 18 days and then anesthetized with 4% isoflurane and killed by cervical dislocation [47]. The number of embryos and implantation sites in the uterus was confirmed by necropsy.

### 2.11. Statistical Analysis

The data are expressed as means ± SDs. Antibody level and testis size of length, width, and testis volume/body weight (g) data from the control and vaccinated groups were compared using the Student’s *t*-test. The normality of the data was evaluated using the Shapiro–Wilk test. Between-group differences in body weight, testis weight, and volume, as well as FSH, LH, and testosterone concentrations were compared through the Mann–Whitney rank sum test. The sperm parameters were compared using Welch’s *t*-test. Statistical analyses were conducted using Sigma Plot Version 14.0 (Systat Software, San Jose, CA, USA). A difference was considered significant at *p* < 0.05.

## 3. Results

### 3.1. Specific Anti-GnRH Immune Response and Serum FSH, LH, and Testosterone Concentrations

The vaccinated mice exhibited GnRH-specific antibody formation in week 5, and this immune response was maintained until week 24 (Figure 2). Control mice did not develop the anti-GnRH antibody. The vaccinated group had a significantly higher antibody level than the control group at any point.

Serum FSH and LH concentrations were measured by day 0 and in weeks 4, 16, and 24 (Figure 3a,b). The FSH and LH concentrations in the vaccinated group decreased gradually after vaccination and were maintained at significantly lower concentrations than those in the control group from weeks 4 to 24 (Figure 3a,b). Serum testosterone concentration (pg/mL) was measured only at week 24. The vaccinated group had a significantly lower concentration (135.3 ± 35.4 pg/mL) than the control group (19,877.4 ± 130 pg/mL). The intra-assay and inter-assay CV% of anti-GnRH IgG and hormone concentration measurements were <10% and 10%, respectively.

### 3.2. Effects of GnRH Vaccine on Testis Weight and Histomorphology

Six male mice in each group were killed in week 24. The vaccinated mice exhibited a significantly lower testis weight (0.033 ± 0.01 g) than the control mice (0.143 ± 0.02 g, *p* = 0.003, Mann–Whitney rank sum test; Table 1). The testis volume (mm^3^) was 76.84 (±11.42) and 380.76 (±60.35) of the vaccinated and control groups, respectively. The testis volume/body weight (g) ratio in the vaccinated and control groups was 1.72 (±0.26) and 7.25 (±1.42), respectively. Furthermore, the testis size of length and width also showed significantly changed in the vaccinated group compared with the control group (Table 1).

Regarding testis histomorphology in the control group, normal structure and the various sperm development stages within the seminiferous tubules were noted (Figure 4a). by contrast, the testes of the vaccinated group exhibited varying degrees of pathologic changes, including the vacuolation of seminiferous tubules and degenerative changes in Leydig cells (Figure 4b). The Leydig cells were edematous, with intracytoplasmic achromatophilic vacuoles and brown pigment accumulation (Figure 4c,d). Germ cell degeneration and depletion were also detected in the vaccinated group, with small numbers of spermatids and sperm observed in the seminiferous tubules (Figure 4e,f).

### 3.3. Cauda Epididymal Sperm Analysis

The analysis of sperm recovered from the cauda epididymis indicated a significantly lower sperm count (1.49 ± 1.1 × 10^6^ vs. 8.1 ± 2.9 × 10^7^ sperm/mL), higher sperm abnormality (75.8% ± 7% vs. 29.3% ± 17%), and lower sperm viability (11.4% ± 11% vs. 43.1% ± 6.5%) in the vaccinated group than in the control group (all *p* < 0.001; Figure 5). Notably, three individuals in the vaccinated group exhibited azoospermia—that is, the absence of measurable sperm in the cauda. A higher percentage of the sperm in the vaccinated group had oval heads and split or divided necks (Figure 5). These results indicated significantly lower sperm quality in the vaccinated group than in the control group.

### 3.4. Mating Behavior and Fertility

Vaginal plug observation revealed that the proportion of mating in the vaccinated group (1/4, 25%) was lower than that in the control group (4/4, 100%). Moreover, the pregnancy ratio in the vaccinated group (1/4, 25%) was lower than that in the control group (3/4, 75%). The average number of pups per litter was 12 and 15 (16/14/15); the average pup weight was 1.312 g and 1.208 ± 0.02 in the vaccinated and control groups, respectively. In the vaccinated group, the same male that mated was with the female that then produced a litter of 12 pups.

## 4. Discussion

Fertility control of free-living animals has become necessary due to the overpopulation of wildlife or feral species [7,48,49]. Numerous population controlling methods have been developed since the 1960s, including steroidal medicine, implants, physical barriers, and surgical castration [48,49]. However, these methods have limitations, such as the need for weekly medicine feeding and animal trapping. In the 2000s, immunocontraception was developed as an alternative [50,51,52]. However, inducing effective immune response in animals with different ages and sexes after immunization without noticeable side effects remains to be challenging.

GnRH, a small endogenous nonimmunogenic decapeptide, plays a vital role in regulating the reproductive system. However, induction of an immune response to self-antigens or acellular antigens is difficult [53,54]. Methods that involve chemical conjugation with carrier proteins and recombinant expression of a fusion protein with danger signals are recommended for improving the immunogenicity of GnRH vaccines [50,55,56]. For both methods, the addition of an adjuvant to the vaccine formulation is suggested to enhance the immune response, but this approach can cause side effects [57]. For example, GonaCon administration in dogs can cause swelling and muscular atrophy at the injection site [32], as well as chronic granulomatous myositis and diffuse coagulative necrosis 33. GonaCon has been documented in a variety of species, and minor to severe side effects were reported in horses (*Equus caballus*), eastern fox squirrels, and non-human primates [58,59]. In the present study, we emulsified a multimer construct protein with an equal volume of MONTANIDE ISA 206 VG adjuvant. The adjuvant comprises a combination of water, oil, and surfactants [60]. It is the most widely used type of adjuvant for animal-use vaccines and is associated with high stability and only mild side effects [61]. At the dosage used in our study, no swelling at the injection site or other side effects were observed after primary immunization and boosters. The specific GnRH antibody level in our study increased after the first boost (week 4), the antibody level remained high for at least 24 weeks with low serum concentrations of reproductive hormones.

In this study, FSH and LH concentrations tended to decline in the vaccinated mice. Immunocontraception GnRH vaccines have been reported to inhibit FHS and LH secretion in gilts [62], heifers [63], and ewes [64]. A study on reproduction in female animals reported a correlation between hormone concentration and ovary size. After GnRH vaccine administration, the weight of the ovary was significantly reduced [65]. The excretion of both FSH and LH in male animals affects spermatogenesis through the regulation of testosterone secretion by Leydig cells [66]. Through the maintenance of the GnRH antibody level, our vaccine reduced the concentration of testosterone, the downstream hormone, and induced testicular atrophy. The vaccinated mice had a significantly lower testis volume and testis volume/body weight ratio than the control group, likely indicating a smaller volume of spermatogenic tissue [67].

Testicular development and function rely on testosterone secretion [68]. Histological evaluation of the mouse testes revealed vacuolation of the seminiferous epithelium and the absence of sperm in the lumen of the vaccinated group. In addition, Leydig cells and germ cells were lower in the vaccinated group than in the control group. These pathologic changes in the testes were also noted in a GnRH antagonist administration test in another study, in which germ cell degeneration and nuclear pyknosis were detected and 80% of tubules were filled with vacuoles [69]. Notably, the brown pigmentation in Leydig cells we observed was also observed in GnRH-deficient mice [70]. Those findings reflect reduced fertility and a cell adhesion defect in the testes attributable to a low concentration of GnRH.

Half of the vaccinated mice (three of six) exhibited azoospermia, and the remaining individuals had lower sperm counts. The sperm of the vaccinated mice also demonstrated a high percentage of oval heads and split necks. Lower sperm quality and higher sperm abnormalities were also found in mice immunized with a GnRH-based vaccine [71]. Leydig cells produce testosterone, which regulates spermatogenesis [72]. The decrease in testosterone levels probably caused sperm abnormalities [73,74]. Sperm count, morphological abnormality, and viability were considered valuable indicators of evaluating fertility and vaccine efficacy [33,75]. The poor semen quality of the vaccinated mice in our study may lead to reduced fertility.

In the pairing test, copulating behavior was observed in mice in both vaccinated and control groups. However, confirmation of mating (vaginal plug observation) and pregnancy rate were reduced in the vaccinated group when compared to the control group. Results revealed that the immunization with the GnRH vaccine was not able to completely suppress the copulating behavior. Studies of other GnRH-based DNA vaccines demonstrated that the pregnancy rate was as high as 50% in vaccinated animals [33,47,76]. However, we found that the concentration of serum testosterone in vaccinated mice was significantly lower than that of the mice in the control group. There was a difference in vaccine reaction between individuals. It is noteworthy to mentioned that one of the immunized mice showing copulating behavior and fertility was the one with the highest concentration of testosterone (990 pg/mL) in the vaccinated group. Testosterone is considered to be the primary hormone for the regulation of sexual behavior in males; nevertheless, other hormones, such as dopamine, glutamate, nitric oxide, and oxytocin can still stimulate sexually excitements [77,78]. The effects of GnRH vaccines on copulating behavior, potential changes in behavior, movement, and activity patterns of animals have not been extensively studied. Quy et al. [79] indicated that no major side effects with respect to activity and movement were found in free-living wild boars (*Sus scrofa*) after the immunization of the GnRH-based vaccine. In addition to the efficacy of fertility control, understanding the influence of immunocontraception on behavior is necessary to ensure the welfare of animals.

The GnRH vaccine is a peptide-based subunit vaccine. In comparison with other modern contraceptive methods, the immunocontraceptive vaccine is safe and easy to produce. However, the poor immunogenicity of the GnRH peptide limits its efficacy and field application. Nevertheless, maintaining prolonged infertility with a single shot of vaccine is more practical for the field. Formulating different adjuvants, in various combinations, to control the release rate of antigens or to elicit desired immune responses maybe able to prolong the duration of infertility in vaccinated animals.

## 5. Conclusions

The efficacy of a recombinant GnRH vaccine in adult male mice was evaluated. This vaccine successfully stimulated the production of IgG specific to GnRH, reduced FSH, LH, and testosterone concentrations in serum, induced testicular atrophy, and reduced sperm quality and fertility without side effects. These findings suggest that our formulation of recombinant GnRH antigen with ISA 206 adjuvant may be able to act as an alternative for controlling overpopulated animals through immunocontraception. However, further research is required to verify its effects in specific species and optimize the procedure for field application.

## Figures and Tables

**Figure 1 vaccines-09-00808-f001:**
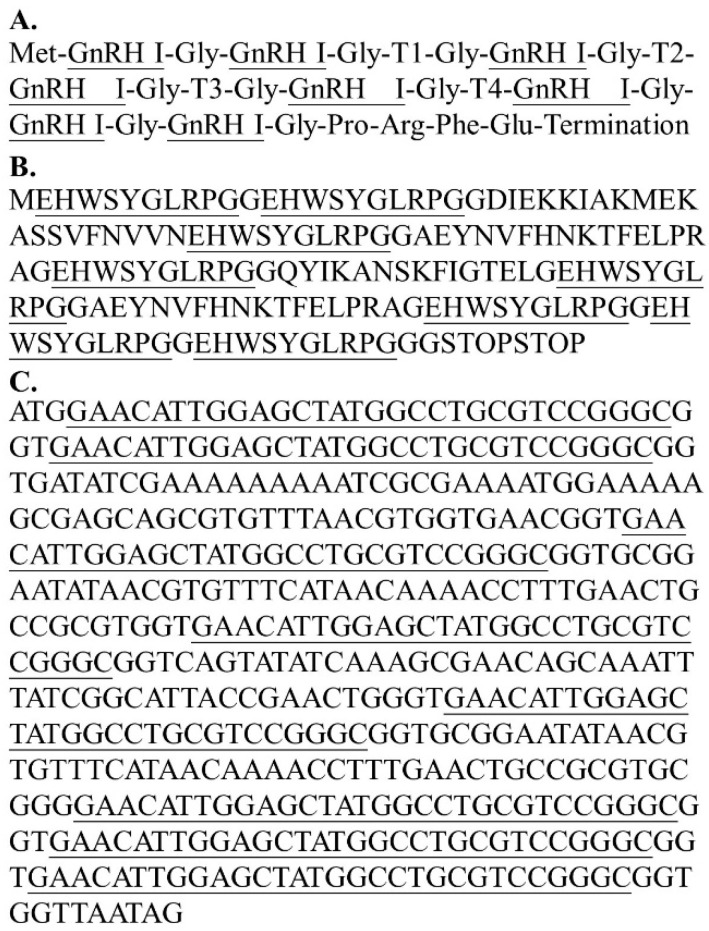
The epitope arrangement, amino acid sequence, and nucleotide sequences of the fusion protein were designed as the gonadotropin-releasing hormone (GnRH) antigen in our GnRH-based vaccine. The design of the fusion protein was modified from that used by Talwar, et al. [37]. (**A**) The multimeric structured fusion protein of the vaccine included eight units of GnRH Ι decapeptide and four small peptides recognized as T-cell epitopes (T1, T2, T3, and T4). (**B**) The amino acid sequence of multimeric GnRH Ι decapeptide linked to T-cell epitopes. (**C**) Nucleotide sequence of multimeric GnRH Ι gene linked to T-cell epitopes.

**Figure 2 vaccines-09-00808-f002:**
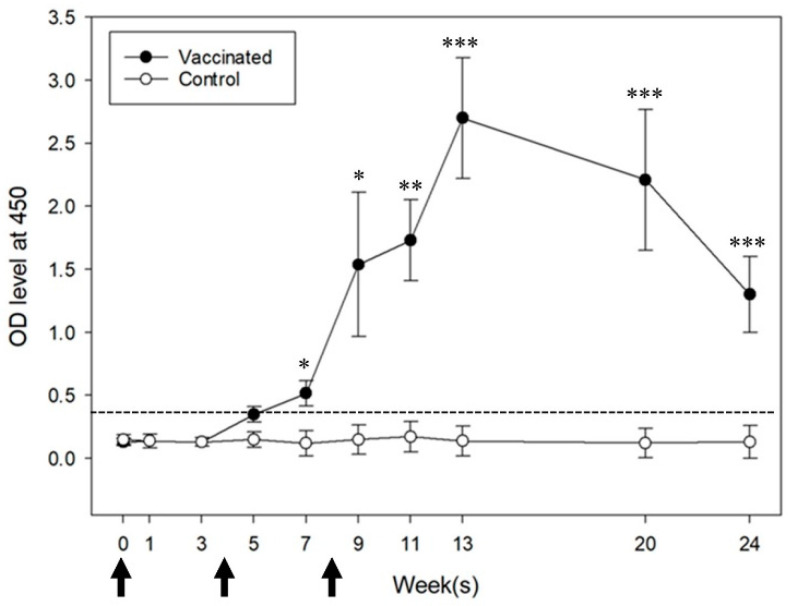
Anti-gonadotropin-releasing hormone (GnRH) IgG antibody response profile (1:50 serum dilution) of the vaccinated and control groups. Arrows on the *x*-axis indicate vaccination schedules in weeks 0, 4, and 8. The dashed line in the figure represents the assay cutoff point 0.384. * *p* < 0.05, ** *p* < 0.01, *** *p* < 0.001 (Student’s *t*-test).

**Figure 3 vaccines-09-00808-f003:**
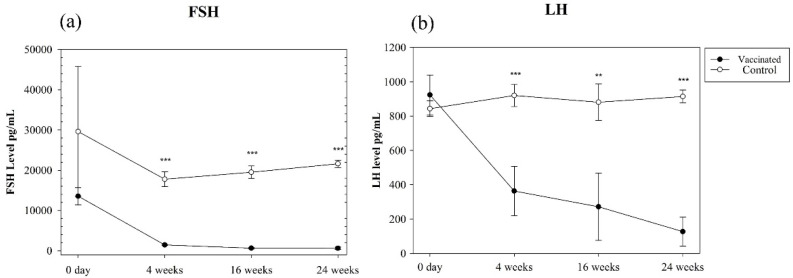
Serum (**a**) follicle-stimulating hormone (FSH) and (**b**) luteinizing hormone (LH) concentrations (pg/mL) in the vaccinated and control groups. The vaccinated group had significantly lower FSH and LH concentrations than the control group. ** *p* < 0.01, *** *p* < 0.001 (Mann–Whitney rank sum test).

**Figure 4 vaccines-09-00808-f004:**
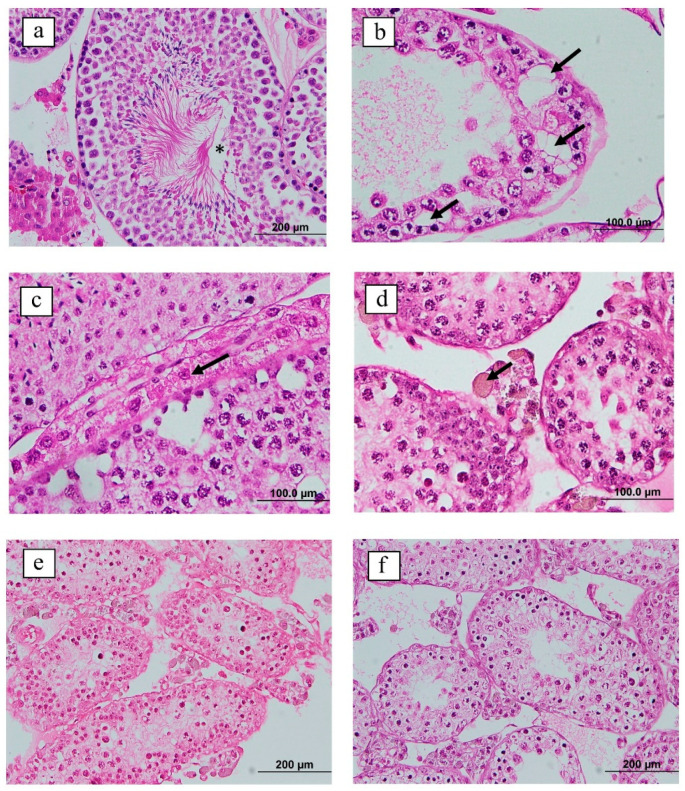
Cross-sections of testis from vaccinated and control mice in week 24. (**a**) Control group: normal testis with normal seminiferous tubules; the lumen (*) is visible, and sperm head coverage is over 80% (hematoxylin and eosin stain, 200×). (**b**) Vaccinated group: multifocal germinal epithelium and Sertoli cell degeneration. Seminiferous tubule vacuolation is seen (arrow; hematoxylin and eosin stain, 400×). (**c**–**f**) Vaccinated group (hematoxylin and eosin stain, 400×) with (**c**) Leydig cells showing gross degenerative changes (arrow); (**d**) brown pigment accumulation in the cytoplasm of Leydig cells (arrow); (**e**) seminiferous tubule degeneration, absence of sperm in the lumen, and germ cell degeneration and depletion; (**f**) disorganized germinal epithelium with few sperm and germ cell degeneration and depletion.

**Figure 5 vaccines-09-00808-f005:**
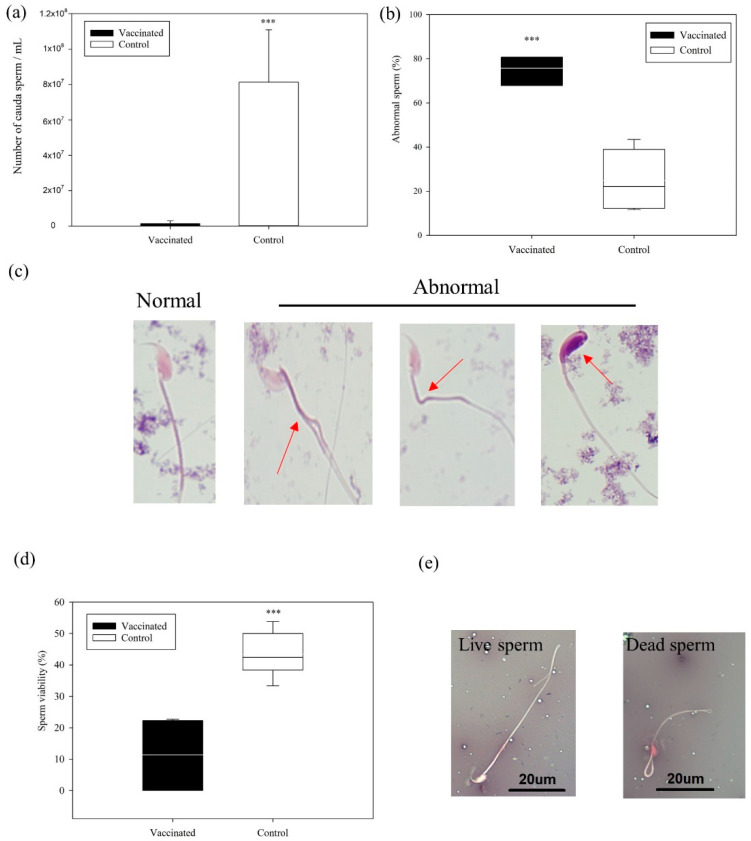
Sperm count, abnormality, and viability in the vaccinated and control groups. (**a**) Sperm count (Mean ± SD). (**b**) Percentage of abnormal sperm in the vaccinated (75.8% ± 7%) and control groups (29.3% ± 17%) (Mean ± SD). (**c**) A high percentage of oval heads and split or divided necks were observed in the sperm in the vaccinated group. (**d**) Percentage of sperm viability in the vaccinated (11.4% ± 11%) and control groups (43.1% ± 6.5%) (Mean ± SD). (**e**) Live and dead sperm identification by eosin–nigrosin staining (*** *p* < 0.001; Welch’s *t*-test).

**Table 1 vaccines-09-00808-t001:** Between-group comparison of testis weight, length, width, volume, body weights, and testis volume/body weight (g) ratio. Testis weight, volume and testis volume/body weight ratio in the vaccinated group was significantly lower than that in the control group (Mann–Whitney rank sum test); testis length, and width also showed significantly lower than the control group (Student’s *t*-test).

	Body Weight (g ± SD)	Testis Weight (g ± SD)	Testis Length (cm ± SD)	Testis Width (cm ± SD)	Testis Volume	Testis Volume/Body Weight (g)
Vaccinated (*n* = 6)	45.18 (±8.3)	0.033 (±0.01)	0.49 (±0.04)	0.34 (±0.07)	76.84 (±11.42)	1.72 (±0.26)
Control (*n* = 6)	53.76 (±8.0)	0.143 (±0.02)	0.90 (±0.48)	0.55 (±0.04)	380.76 (±60.35)	7.24 (±1.42)
*p*-value	*p* = 0.132	*p* = 0.003	*p* < 0.001	*p* < 0.001	*p* < 0.001	*p* = 0.002

## Data Availability

The data presented in this study are contained within the article.

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
