# Peer review of "Effects of a Recombinant Gonadotropin-Releasing Hormone Vaccine on Reproductive Function in Adult Male ICR Mice"

_vaccines, 2021, doi:10.3390/vaccines9080808_

Round 1

Reviewer 1 Report

Recombinant GnRH vaccine in male mice.

This manuscript describes a small study of the effects of a recombinant GnRH -multimeric fusion protein given 3 x times intramuscularly to male ICR mice.  Responses in terms of antibodies, endocrine hormones, sperm parameters and reproductive organs of control (n=10) and treated (n=10) mice were compared.  Pairings of treated males with untreated females (n=4 pairs only) indicated some males were infertile.

The study itself is well described and the findings are adequately explained.  However, the background information in the Introduction in relation to management of wildlife is limited and could be more comprehensive.

The conclusion that this vaccine could be used for population of free-living mammals is rather naïve - - the authors give no consideration to practical and cost-effective delivery - for example oral delivery is not mentioned?  How would you ensure free-living animals received primary and booster treatments?

There is some improvement of English expression required.

Specific comments

Abstract

Line 20: Change to read as “We evaluated a .... with the adjuvant MONTANIDE ISA 206 VG as a GnRH vaccine in...”

Line 24:  Change to read as “immune response by week 5...”

Line 25:  Comment “maintained the specific antibody for at least 24 weeks.  Perhaps a clearer statement would indicate that peak antibody levels were observed after the second boost by 13 weeks, and antibodies were elevated, but declining by 24 weeks.

Line 26:  Delete “In addition”

Line 27:  Add after viability “compared to control males”.

Line 27:  “The fertility test....”  This is an unclear sentence as “fertility test” is not defined.  Perhaps rewrite as:  “The outcomes of pairings of treated males with untreated females revealed reduced matings and number of litters compared to control pairings.

Lines 28-30:  These concluding sentences represent premature statements as to the feasibility of using the vaccine in the field (see above comments).

Line 32:  Keywords:  Change “animal” to “mammal”

Introduction

Several references should be considered for inclusion in the Introduction (Lines 34-45) and/or Discussion

Bomford, M. (1990). A role for fertility control in wildlife management? Australian Government Publishing Service, Canberra.

Bomford, M., and O'Brien, P. (1992). A role for fertility control wildlife management in Australia? In ‘Proceedings of the Fifteenth Vertebrate Pest Conference’. (Ed. J. E. Borrecco and R. E. Marsh) pp. 344-347. (University of California: Davis, CA.)

Fagerstone et al (2008)  Registration of wildlife contraceptives in the USA….  Wildlife Research 2008, 35, 586-592.

Fagerstone, K. A., Miller, L. A., Killian, G., and Yoder, C. A. (2010). Review of issues concerning the use of reproductive inhibitors, with particular emphasis on resolving human‐wildlife conflicts in North America. Integrative Zoology 5, 15-30.

Hampton, J. O., Hyndman, T. H., Barnes, A., and Collins, T. (2015). Is wildlife fertility control always humane? Animals 5, 1047-1071.

Miller, L.A., Johns, B.E. and Killian, G.J. (2000). Immunotcontraception of White-tailed deer with GnRH vaccine.  American Journal of Reproductive Immunology, 44, 266-274.

Massei, G., and Cowan, D. (2014). Fertility control to mitigate human–wildlife conflicts: a review. Wildlife Research 41, 1-21.

Miller, L. A., Fagerstone, K. A., and Eckery, D. C. (2013). Twenty years of immunocontraceptive research: lessons learned. Journal of Zoo and Wildlife Medicine 44, S84-S96.

Pepin, K. M., Davis, A. J., Cunningham, F. L., VerCauteren, K. C., and Eckery, D. C. (2017). Potential effects of incorporating fertility control into typical culling regimes in wild pig populations. PloS one, 12(8), e0183441.

Sharma, S., and Hinds, L. A. (2012). Formulation and delivery of vaccines: ongoing challenges for animal management. Journal of Pharmacy and Bioallied Sciences 4, 258-266.

Line 50:  Reference 8 is about GnRH receptors not GnRH - relevance here?

Line 53:  Male immuncontraception or immunocastration may alter behaviour and impact welfare of animals - for example GnRH vaccinate white-tailed deer (Miller et al 2000) in which antlers do not harden off can lead to adverse impacts.  Targetting females alone is a better proposition.

Line 64;  As well as reference 12, also quote Fagerstone et al 2008?

Line 75:  Do the authors mean “sexually immature” animals here”?  Many studies have shown efficacy of GnRH-based vaccines in mature adults - see review by Massei and Cowan (2014).

Line 82:  Change “fertility” to “outcomes of pairings with untreated females”.

Line 85:  Please provide a justification for the use of 30 week-old mice?  These would be considered old mice in a field situation. At the time of pairing animals were 54 weeks old. Most lab-based stuides of fertiity control agents in rodents use mice that are 6-12 weeks old during the experimetnal period.  

Line 97:  Please describe the modification of the conjugation of the fusion protein compared to that used by Talwar et al.

Lines 98-99 and Line 104 onwards:  If the 0.2 ml/200ug antigen was emulsified with an equal volume of adjuvant, did an animal receive 0.4ml for each intramuscular injection.  If only 200ul was injected then the animals only received 100ug antigent for each dose?

Was the site of injection the hindquarters (rump) of the mouse?  Same side or alternating sides for the booster treatments?  Was the volume of PBS plus adjuvant also 400ul for control animals?

Line 118:  Please indicate here that only the cauda epididymis was used for assessment of sperm.

Line 144:  Please note that OD values are not interchangeable with titer values.  Unless dilutions of serum were done to calculate titers authors should be careful how they express their findings.

Line 175:  After ficxation in Bouin’s for 24h presumably the tissues were wshed and then passed through graded alcohols before embedding in paraffin?

Line 205:  Change to Wyrobek et al (1983).

Lines 209 onwards:  A very small sample size (n=4 pairs) was used for this assessment of male fertility. Justify?

Line 214:  Change to read as:  “If a plug was observed, the female mouse was housed individually for 18 days and then anesthetized with 4% isoflurane. It was killed by cervical dislocation tand dissected to confirm pregnancy and determine the number of embryos and implantation sites in the uterus.”

Line 240:  Change to “….was measured only at Week 24.”

Line 245 onwards:  Comment on lower body weight in vaccinated mice - were body weights the same at the start of the experiment?  Also testis weight per g body weight is useful.  However, for future studies a more precise measure of any treatment changes is derived if testis volumes are calculated - use vernier calipers to measure length and width of dissected testis.

Figure 2:  Note earlier comment about OD values versus antibody titers.  Values are Mean ± SD?

Figure 3:  The quality of theis figure  could be imprroved by enlarging font size and thickness.

Figure 4:  Is this figure warranted?  The values are stated in the text at Line 241.  Delete?

Figure 5:  Useful and informative.

Figure 6: Explain plots a, b and d, more precisely - whisker plots?  Median values; Confidence intercals? Mean ± SD?

Table 2:  Probably not needed?  Delete.  Description in text could be sufficient.  Please clarify in text whether the same male that mated was with the female that then produced a litter of 12 pups?  Also include this litter size and weight information in the text along with information for the control litters.

Discussion  - general comments for consideration

Many of the paragraphs re-describe the results then make a statement with respect to the literature.  This is not a very strong contextual discussion. And some issues are not addressed - for example delivery to wild populations is a major problem for immunocontraceptive vaccines (see overview comments above)

Paragraph 2: Line 325:  side effects are seen in dogs after GonaCon but not in majority of other species - see review by Massei and Cowan (2014). Perhaps a different adjuvant mught be more useful for dogs?

Concluding paragraph:  As noted above practical delivery to wild populations is not addressed - need a single orally delivered effective treatment.  There are many more steps between this first set of lab results  using a small sample size to effective field delivery.  

Author Response

Dear reviewer 1,

We appreciate your valuable comment. Those insightful comments led to great improvements to our manuscript. The authors have carefully considered all the comments and tried their best to do revisions.

We explain point-by-point the details of the comments. All modifications in this article had been using the "Track Changes" function in Microsoft Word. We appreciate the time and effort of your help.

Response to Reviewer 1 Comments
Dear reviewer 1,
We appreciate your valuable comment. Those insightful comments led to great 
improvements to our manuscript. The authors have carefully considered all the comments and 
tried their best to do revisions. 
We explain point-by-point the details of the comments. All modifications in this 
article had been using the "Track Changes" function in Microsoft Word. We appreciate the 
time and effort of your help.
Sincerely,
Ai-Mei Chang
Overview comments:
Point 1: This manuscript describes a small study of the effects of a recombinant GnRH -
multimeric fusion protein given 3 x times intramuscularly to male ICR mice. Responses in 
terms of antibodies, endocrine hormones, sperm parameters and reproductive organs of 
control (n=10) and treated (n=10) mice were compared. Pairings of treated males with 
untreated females (n=4 pairs only) indicated some males were infertile.
The study itself is well described and the findings are adequately explained. 
Response 1: We deeply appreciate the reviewer’s comment and suggestion to improve the 
quality of our manuscript.
Point 2: However, the background information in the Introduction in relation to management 
of wildlife is limited and could be more comprehensive.
Response 2: Thank you for the concern. More descriptions regarding overpopulated freeliving wildlife, conflict, and management methods were added (line 44-58). 
Point 3: The conclusion that this vaccine could be used for population of free-living 
mammals is rather naïve - - the authors give no consideration to practical and cost-effective 
delivery - for example oral delivery is not mentioned? How would you ensure free-living 
animals received primary and booster treatments?Response 3: We agreed that there are still many barriers that need to be overcome before 
the vaccine can be used to control the population of free-living animals. We have tried to 
have the vaccine orally administered to mice through nano-peptide and lipo-peptide 
carriers without success. Apparently, the oral delivered vaccine was not able to induce 
sufficient immune responses in the gastrointestinal (GI) tract, so an increase in antibody 
against GnRH in serum was not observed. Therefore, injection-based vaccination is still 
more particle. The remote wildlife vaccine delivery system can be a new option for vaccine 
delivery for free-living animals in the future. This system is equipped with a Hub with bait 
to attract targeted animals, which can be vaccinated via a low-velocity dart. When used in 
combination with auto-camera, RFID, and AI facial recognition, this would be a feasible 
approach for controlling the population of free-living animals.
Point 4: There is some improvement of English expression required.
Response 4: Thanks for the reviewer's suggestion. We had polished the language in this 
revision.
Specific comments
Point 1: Line 20: Change to read as “We evaluated a .... with the adjuvant MONTANIDE 
ISA 206 VG as a GnRH vaccine in...
Response 1: Thank you, we revised the sentence at lines 20-21. 
Point 2: Line 24: Change to read as “immune response by week 5...”
Response 2: Thank you, we revised the sentence at line 25.
Point 3: Line 25: Comment “maintained the specific antibody for at least 24 weeks. Perhaps 
a clearer statement would indicate that peak antibody levels were observed after the second 
boost by 13 weeks, and antibodies were elevated, but declining by 24 weeks.
Response 3: Thanks for the reviewer’s comment and point out our deficiency. To clarify 
the dynamic change of antibody level, we rewrote the sentence at lines 25-26. The 
sentence was changed to “The vaccine induced a specific immune response by week 5, the 
peak of antibody level was showed by week 13 and declined the specific antibody 
gradually until the end of the study (week 24)”.
Point 4: Line 26: Delete “In addition”
Response 4: Thank you for the suggestion, we deleted “In addition” in the sentence.
Point 5: Line 27: Add after viability “compared to control males”.
Response 5: Thank you for the suggestion, we added “compared to control males” after 
viability at line 29.Point 6: Line 27: “The fertility test....” This is an unclear sentence as “fertility test” is not 
defined. Perhaps rewrite as: “The outcomes of pairings of treated males with untreated 
females revealed reduced matings and number of litters compared to control pairings.
Response 6: Thank you for the suggestion The sentence was revised to “The outcomes of 
pairings of treated males with untreated females revealed reduced mating, pregnancy rates 
and the number of litters compared to control pairings.” (lines 30-31).
Point 7: Lines 28-30: These concluding sentences represent premature statements as to the 
feasibility of using the vaccine in the field (see above comments).
Response 7: Thank you for the suggestion. We revisited the conclusion of the abstract;
the limitation and the future work were mentioned at lines 32-34.
Point 8: Line 32: Keywords: Change “animal” to “mammal”
Response 8: We revised the whole sentence to “The immune responses elicited by the 
same vaccine should not be similar in different species of animals. Therefore, 
accumulating more research data on using GnRH vaccines in different species is 
fundamental and needed for optimizing the efficacy and application”.
Point 9:
Several references should be considered for inclusion in the Introduction (Lines 34-45) 
and/or Discussion
Bomford, M. (1990). A role for fertility control in wildlife management? Australian 
Government Publishing Service, Canberra.
Bomford, M., and O'Brien, P. (1992). A role for fertility control wildlife management in 
Australia? In ‘Proceedings of the Fifteenth Vertebrate Pest Conference’. (Ed. J. E. Borrecco 
and R. E. Marsh) pp. 344-347. (University of California: Davis, CA.)
Fagerstone et al (2008) Registration of wildlife contraceptives in the USA…. Wildlife 
Research 2008, 35, 586-592.
Fagerstone, K. A., Miller, L. A., Killian, G., and Yoder, C. A. (2010). Review of issues 
concerning the use of reproductive inhibitors, with particular emphasis on resolving human‐
wildlife conflicts in North America. Integrative Zoology 5, 15-30.Hampton, J. O., Hyndman, T. H., Barnes, A., and Collins, T. (2015). Is wildlife fertility 
control always humane? Animals 5, 1047-1071.
Miller, L.A., Johns, B.E. and Killian, G.J. (2000). Immunotcontraception of White-tailed deer 
with GnRH vaccine. American Journal of Reproductive Immunology, 44, 266-274.
Massei, G., and Cowan, D. (2014). Fertility control to mitigate human–wildlife conflicts: a 
review. Wildlife Research 41, 1-21.
Miller, L. A., Fagerstone, K. A., and Eckery, D. C. (2013). Twenty years of 
immunocontraceptive research: lessons learned. Journal of Zoo and Wildlife Medicine 44, 
S84-S96.
Pepin, K. M., Davis, A. J., Cunningham, F. L., VerCauteren, K. C., and Eckery, D. C. (2017). 
Potential effects of incorporating fertility control into typical culling regimes in wild pig 
populations. PloS one, 12(8), e0183441.
Sharma, S., and Hinds, L. A. (2012). Formulation and delivery of vaccines: ongoing 
challenges for animal management. Journal of Pharmacy and Bioallied Sciences 4, 258-266.
Response 9: Thank you for the recommendation, we read through those references and 
incorporated the information, and cited the literature in our manuscript.
Point 10: Line 50: Reference 8 is about GnRH receptors not GnRH - relevance here?
Response 10: This citation was removed.
Point 11: Line 53: Male immuncontraception or immunocastration may alter behaviour and 
impact welfare of animals - for example GnRH vaccinate white-tailed deer (Miller et al 2000) 
in which antlers do not harden off can lead to adverse impacts. Targetting females alone is a 
better proposition.
Response 11: Thank you for the concern. Targeting on females of deer would indeed be a 
better proposition. However, in other species, according to our observation, after 
vaccination the male mice individual could show mating behavior, mounting. Although 
testosterone regulates the sexual behavior of males, other hormones, such as dopamine, 
glutamate, nitric oxide, and oxytocin could stimulate the sexually excitatory of an 
individual [1]. We added this description in lines 423-433.We agreed with your concern about animal welfare in male deer. It needs to be more 
consideration if using the GnRH vaccine in male deer. However, the possible side effect of 
the GnRH vaccine on different species might exhibit differently. Therefore, as we 
mentioned that the vaccine should be evaluated for the efficiency and possible side effects
before deployment for specific species. One study by Quy et al. [2] showed that after the 
treatment of the GnRH vaccine, no major side effects on activity and movement were 
found. We added this description at line 428.
The influence of such vaccines on animal welfare requires more investigations.
Point 12: Line 64; As well as reference 12, also quote Fagerstone et al 2008?
Response 12: We had cited this reference at line 82.
Point 13: Line 75: Do the authors mean “sexually immature” animals here”? Many studies 
have shown efficacy of GnRH-based vaccines in mature adults - see review by Massei and 
Cowan (2014).
Response 13: To clarify the meaning, in this paragraph we mean the injection time of the 
vaccine. Most studies conducted the vaccination before peripubertal phase, when 
reproductive tissues haven’t matured (sexually immature animal). According to previous 
references, we already knew the GnRH vaccine can “inhibit” the growth of reproductive 
tissue, but scarce study work on the effect of GnRH vaccine on the matured individuals. In 
this study, we immunized 30-week-old mice (middle-aged) and noticed the atrophy of 
testis. We revised the manuscript at lines 94-97. Thanks to the reviewer to point out the 
vague description.
Point 14: Line 82: Change “fertility” to “outcomes of pairings with untreated females”.
Response 14: Thank you for the suggestion, we revised the sentence “fertility” to 
“outcomes of pairings with untreated females” at line 104.
Point 15: Line 85: Please provide a justification for the use of 30 week-old mice? These 
would be considered old mice in a field situation. At the time of pairing animals were 54 
weeks old. Most lab-based stuides of fertiity control agents in rodents use mice that are 6-12 
weeks old during the experimetnal period.
Response 15: Thank you for the concern. According to the life phase equivalence of mice 
and humans, the 30-week-old mice equal to the human around 30 years old [3]. The 
previous study had proved the effectiveness of the GnRH vaccine administrated in 
peripubertal individuals. Because we aimed to know the effect of GnRH vaccine on
middle-aged individuals. We revised this sentence at lines 108-110 in the manuscript. 
Therefore, we choose the 30-week-old mice as experimental animals. In our research, the control group was at the same age, their mating rate (100%) and pregnancy rate (75%)
were higher than the vaccinated group and had a similar result with other studies
(pregnancy rate : 80%) [4]. We had already evaluated the effect of this GnRH vaccine
vaccinated at 6 weeks old individuals (data not shown) with a similar result in another 
experiment. We hoped this research can provide useful information for future study.
Point 16: Line 97: Please describe the modification of the conjugation of the fusion protein 
compared to that used by Talwar et al.
Response 16: We added a paragraph to describe the detail of modification at line 123-124.
The modification was repeated more GnRH in fusion protein and choose Escherichia coli
as the expression system of the fusion protein.
Point 17: Lines 98-99 and Line 104 onwards: If the 0.2 ml/200ug antigen was emulsified 
with an equal volume of adjuvant, did an animal receive 0.4ml for each intramuscular 
injection. If only 200ul was injected then the animals only received 100ug antigent for each
dose?
Was the site of injection the hindquarters (rump) of the mouse? Same side or alternating 
sides for the booster treatments? Was the volume of PBS plus adjuvant also 400ul for control 
animals?
Response 17: Thanks for the reviewer’s comment and point out the vague statement. We 
rewrote the paragraph on vaccine preparation and Immunization in M&M at lines 124-127
and 131-132. In the vaccine preparation, the concentration of stock antigen was 2mg/mL.
After diluted to 1 mg/mL with oil-based MONTANIDE ISA 206 VG adjuvant and 0.2 mL 
(200 μg) were used for vaccination.
The vaccine was intramuscularly injected in right and left hindquarters for 0.1mL each site
of the mice and was boost at the same position. The volume of PBS plus adjuvant was 0.2 
mL, the procedure was the same as the vaccination group.
Point 18: Line 118: Please indicate here that only the cauda epididymis was used for 
assessment of sperm.
Response 18: We revised the sentence here. The sperm was collected from “cauda” part of 
the epididymis.
Point 19: Line 144: Please note that OD values are not interchangeable with titer values. 
Unless dilutions of serum were done to calculate titers authors should be careful how they 
express their findings.Response 19: Thanks for the comment and point out the vague description. We replaced 
the word “antibody titer” by “antibody level” in the manuscript. We hoped this description 
can clarify our expression.
Point 20: Line 175: After ficxation in Bouin’s for 24h presumably the tissues were wshed 
and then passed through graded alcohols before embedding in paraffin?
Response 20: Yes, after fixation, the tissues were washed and then passed through graded 
alcohols before embedding in paraffin. We added the detail of this description in the 
manuscript at lines 215-216. 
Point 21: Line 205: Change to Wyrobek et al (1983).
Response 21: We revised the citation according to the guideline of “Reference List and 
Citations Style Guide for MDPI Journals” at line 246.
Point 22: Lines 209 onwards: A very small sample size (n=4 pairs) was used for this 
assessment of male fertility. Justify?
Response 22: Thank you for the reviewer’s concern. We followed the reproductive study 
reference for the mating test by Saucedo et al. [5]. In that study, they compared the 
reproductive ability of fibroblast growth factor 2 deficiency mice and wild-type mice 
(control group), the number of mice in each group was 3-5. Because we need to sacrificed 
part of the animal before the mating test, we chose the median number 4 for the matting 
test. 
Point 23: Line 214: Change to read as: “If a plug was observed, the female mouse was 
housed individually for 18 days and then anesthetized with 4% isoflurane. It was killed by 
cervical dislocation tand dissected to confirm pregnancy and determine the number of 
embryos and implantation sites in the uterus.”
Response 23: Thank you for the suggestion. We revised the sentence to clarify our mating 
test. The revision was at lines 254-258.
Point 24: Line 240: Change to “….was measured only at Week 24.”
Response 24: We revised the sentence to “Serum testosterone concentration (pg/mL) was 
detected measured only at week 24” (line 283).
Point 25: Line 245 onwards: Comment on lower body weight in vaccinated mice - were 
body weights the same at the start of the experiment? Also testis weight per g body weight is 
useful. However, for future studies a more precise measure of any treatment changes is 
derived if testis volumes are calculated - use vernier calipers to measure length and width of 
dissected testis.Response 25: Thank you for the comment. We recorded the weight changed every 2 
weeks. Before the experiment, the average weight (Mean ± SD) of the vaccinated and 
control group was 38.69±1.14 and 39.7±1.4, respectively. From the beginning to the end 
of the experiment, there were no significant differences between the body weight of each 
group. For the testis length and width, as you mentioned, we did measure after sacrificed 
animals. We added this information in Table1 and revised the manuscript at lines 291-293.
Point 26: Figure 2: Note earlier comment about OD values versus antibody titers. Values 
are Mean ± SD?
Response 26: Thank you for the comment. The OD values are Mean ± SD. We revised all 
the “antibody titer” to “antibody level” to clarify the meaning. 
Point 27: Figure 3: The quality of theis figure could be imprroved by enlarging font size 
and thickness.
Response 27: Thank you for the suggestion, we agreed with your comment and improve 
the quality of the figure. The font size and thickness were modified. The revised figure 
was showed under this response and at line 317 in the manuscript.
Point 28: Figure 4: Is this figure warranted? The values are stated in the text at Line 241. 
Delete?
Response 28: The figure of testosterone level was removed from the manuscript as the 
result being described in the main text.
Point 29: Figure 5: Useful and informative.
Response 29: Thank you for the comment.
Point 30: Figure 6: Explain plots a, b and d, more precisely - whisker plots? Median values; 
Confidence intercals? Mean ± SD?
Response 30: Thank you for the suggestion, the numbers in this figure were Mean ± SD.
We added the description in this figure.Point 31: Table 2: Probably not needed? Delete. Description in text could be sufficient. 
Please clarify in text whether the same male that mated was with the female that then 
produced a litter of 12 pups? Also include this litter size and weight information in the text 
along with information for the control litters.
Response 31: Thank you for the suggestion. We deleted Table2 and added more 
descriptions. We included litter size and weight information in the text of the vaccinated 
and control group at lines 354-357.
Point 32:
Many of the paragraphs re-describe the results then make a statement with respect to the 
literature. This is not a very strong contextual discussion. And some issues are not addressed 
- for example delivery to wild populations is a major problem for immunocontraceptive 
vaccines (see overview comments above)
Response 32: Thank you for the concern. We agreed with you and had incorporated this 
suggestion throughout our paper. In this revision, we introduced more background of 
animal management method at line 44-57 and discussed the behavior observation after 
vaccination at line 423-433. 
For the issue “delivery to wild populations”, we agreed that this is an important concern 
before field application. However, the main concern in this manuscript is the efficacy of 
the recombinant GnRH on reproductive function in adult male ICR mice, but not the 
delivery system. As we replied to Response 3, the remote wildlife vaccine delivery system 
can be a new option for vaccine delivery at the free-living animal in the future. However, 
the delivery system needs further evaluation in specific species.
Point 33: Paragraph 2: Line 325: side effects are seen in dogs after GonaCon but not in 
majority of other species - see review by Massei and Cowan (2014). Perhaps a different 
adjuvant mught be more useful for dogs?
Response 33: Thank you for the comment. GonaCon has been documented in a variety of 
species, and minor to severe side effects were reported in horses (Equus caballus), eastern 
fox squirrels, and non-human primates [6,7]. We had the sentence to describe the side effect 
of GonaCon in other species at lines 377-379. Based on the discussion with researchers in 
USDA, the adjuvant in GonaCon had severe side effects on non-human primates which is 
one of our future target species (Macaca cyclopis). However, Montanide ISA206 (waterin–oil-water emulsions, W/O/W) is the commercial adjuvant. This adjuvant has been used 
to emulsify with varied antigens, and no obvious negative side effect was noticed. We 
thought this adjuvant will be useful for canine and other species.Point 34:
Concluding paragraph: As noted above practical delivery to wild populations is not 
addressed - need a single orally delivered effective treatment. There are many more steps 
between this first set of lab results using a small sample size to effective field delivery. 
Response 34: Thank you for the concern. We understand the difficulty from lab to field. 
In this research, we developed and evaluated a recombinant GnRH vaccine designed and 
produced by our lab. This vaccine can successfully stimulate GnRH IgG antibodies; reduce 
FSH, LH, and testosterone concentrations; induce testicular atrophy, and reduce sperm 
quality and fertility.
The future evaluated in target species should be evaluated, due to the species-specific
reaction and different reproductive cycles. For oral vaccine administration, we did try to 
develop the oral GnRH vaccine in other experiments. However, it is a great challenge for 
peptide vaccines, like the GnRH vaccine, to overcome the harsh gastrointestinal (GI) 
environment. We did not have a good result of the oral GnRH vaccine. Based on our 
experience, a DNA vaccine or vector vaccine could be an option for oral GnRH vaccine 
design to solve this problem. However, it is out of the scope of this manuscript, and 
therefore, we did not describe the oral vaccine in the manuscript. We hope that the Editor 
and reviewer can understand and accept our point of view.
1. Hull, E.M.; Meisel, R.L.; Sachs, B.D. Male sexual behavior. In Hormones, brain and 
behavior, Elsevier: 2002; pp. 3-137.
2. Quy, R.J.; Massei, G.; Lambert, M.S.; Coats, J.; Miller, L.A.; Cowan, D.P. Effects of a 
GnRH vaccine on the movement and activity of free-living wild boar (Sus scrofa). 
Wildlife Research 2014, 41, 185-193.
3. Fox, J.G.; Barthold, S.; Davisson, M.; Newcomer, C.E.; Quimby, F.W.; Smith, A. The 
mouse in biomedical research: normative biology, husbandry, and models; Elsevier: 
2006; Vol. 3.
4. Chua, B.Y.; Zeng, W.; Lau, Y.F.; Jackson, D.C. Comparison of lipopeptide-based 
immunocontraceptive vaccines containing different lipid groups. Vaccine 2007, 25, 
92-101.
5. Saucedo, L.; Rumpel, R.; Sobarzo, C.; Schreiner, D.; Brandes, G.; Lustig, L.; Vazquez‐
Levin, M.H.; Grothe, C.; Marín‐Briggiler, C. Deficiency of fibroblast growth factor 2 
(FGF‐2) leads to abnormal spermatogenesis and altered sperm physiology. Journal 
of cellular physiology 2018, 233, 9640-9651.
6. Krause, S.K.; Van Vuren, D.H.; Laursen, C.; Kelt, D.A. Behavioral effects of an 
immunocontraceptive vaccine on eastern fox squirrels. The Journal of Wildlife 
Management 2015, 79, 1255-1263.7. Baker, D.L.; Powers, J.G.; Ransom, J.I.; McCann, B.E.; Oehler, M.W.; Bruemmer, J.E.; 
Galloway, N.L.; Eckery, D.C.; Nett, T.M. Reimmunization increases contraceptive 
effectiveness of gonadotropin-releasing hormone vaccine (GonaCon-Equine) in freeranging horses (Equus caballus): Limitations and side effects. PLoS One 2018, 13, 
e0201570

Please see the attachment for the response.

Sincerely,

Ai-Mei Chang

Reviewer 2 Report

Review of Chang et al “Effects of a recombinant gonadotropin-releasing hormone vaccine on reproductive function in adult male ICR mice”

Vaccines 2021

In this manuscript authors have evaluated the effects of an immune-contraceptive vaccine against the Gonadotropin-releasing hormone prepared using GnRH fusion protein with MONTANIDE ISA 206 VG adjuvant in adult ICR mice. Authors show that the vaccine induced immune response in form of anti-GnRH specific antibodies which lasted for at least 24 weeks. Vaccine administration also reduced the levels of hormones like testosterone, follicle stimulating hormone and luteinizing hormone, reduced testis size and histomorphology, and detrimentally affected sperm quality, concentration, morphology and viability. Authors also showed that the vaccine also reduced mating and pregnancy rates.

Here, authors have demonstrated using a large spectrum of tests, that the vaccine afflicts the reproductive capacity in mice. The manuscript is written clearly. Some minor issues are stated below. The data is presented well and convincing.

However, the study generated no data about the side effects of the vaccines. Since the authors suggest that current methods of controlling te stray animal population are inefficient and or inhumane, it is important to show that the anti-GnRH vaccine used in this study does not cause life threatening complications and does not affect the vital body functions in the test animals. The effects of this contraceptive vaccine on female mice need to be studied more.Moreover, vaccines against GnRH have been successfully used in horses, elks and pigs therefore there is little novelty to this work. Additionally, previous anti-GnRH vaccines required multiple, which made the vaccination cumbersome and difficult to achieve. How do the authors address this concern?

Specific comments:

Line 21: Define ICR mice.

Line 50: Remove theoretically.

Line 55: Rewrite this line “In general, GnRH is highly conserved and typically exhibits an identical structure in phylogenetically closed species”

Line 78: Define ISA206.

Line 81: sex hormones not sexual hormones

Line122: no structure is shown in the figure. The line reads obscurely

Line 167: It is not clear what intra and interassays mean.

Line 236: on day 0 not in day 0

Line 242: Please explain what intra and interassays are.

What are the effects of this contraceptive vaccine on female mice?

Line 332: “In our study” not “Im our study”

Author Response

Dear reviewer 2, 

We appreciate your valuable comment. Those insightful comments led to great improvements to our manuscript. The authors have carefully considered all the comments and tried their best to do revisions. We hope the revision can meet your standard.  
We explain point-by-point the details of the comments. All modifications in this article had been using the "Track Changes" function in Microsoft Word. We appreciate the time and effort of your help.

Response to Reviewer 2 Comments
Dear reviewer 2,
We appreciate your valuable comment. Those insightful comments led to great 
improvements to our manuscript. The authors have carefully considered all the comments and 
tried their best to do revisions. We hope the revision can meet your standard. 
We explain point-by-point the details of the comments. All modifications in this 
article had been using the "Track Changes" function in Microsoft Word. We appreciate the 
time and effort of your help.
Sincerely,
Ai-Mei Chang
Overview comments:
Point 1: In this manuscript authors have evaluated the effects of an immune-contraceptive 
vaccine against the Gonadotropin-releasing hormone prepared using GnRH fusion protein 
with MONTANIDE ISA 206 VG adjuvant in adult ICR mice. Authors show that the vaccine 
induced immune response in form of anti-GnRH specific antibodies which lasted for at least 
24 weeks. Vaccine administration also reduced the levels of hormones like testosterone, 
follicle stimulating hormone and luteinizing hormone, reduced testis size and 
histomorphology, and detrimentally affected sperm quality, concentration, morphology and 
viability. Authors also showed that the vaccine also reduced mating and pregnancy rates.
Here, authors have demonstrated using a large spectrum of tests, that the vaccine afflicts the 
reproductive capacity in mice. The manuscript is written clearly. Some minor issues are 
stated below. The data is presented well and convincing.
Response 1: We deeply appreciate the reviewer’s comment and suggestion to improve the 
quality of our manuscript.
Point 2: However, the study generated no data about the side effects of the vaccines. Since 
the authors suggest that current methods of controlling te stray animal population are 
inefficient and or inhumane, it is important to show that the anti-GnRH vaccine used in this study does not cause life threatening complications and does not affect the vital body 
functions in the test animals. 
Response 2: Indeed, it is very important to prove the safety of a vaccine. Since we are at 
the early stage of vaccine development so that we were more focusing on the 
immunogenicity of the GnRH vaccine. Nevertheless, we did record the body weight of 
each individual, there is no difference between vaccination and control group. Otherwise,
there were no clinical threatening complications after injection. During prime and booster 
doses, no swelling at the injection site swelling or other side effects were observed. (line 
385-387) We will have a more detailed evaluation of the safety of the vaccine in 
immunized animals in our future studies. 
Point 3: The effects of this contraceptive vaccine on female mice need to be studied more. 
Response 3: Thank you for the comment. We also tested the effects of this vaccine in 
female mice. Results demonstrated that immunizing female mice with our 
immunocontraceptive vaccine had decreased sex hormones, including, progesterone and 
estrogen. More experiments and data are required to determine whether vaccination can 
prevent or reduce estrous cycles and ovulation. However, this manuscript was focus on 
the effect of the GnRH vaccine on male mice and therefore, we did not include the result 
of female mice experiment.
Point 4: Moreover, vaccines against GnRH have been successfully used in horses, elks and 
pigs therefore there is little novelty to this work.
Response 4: We also tried the GnRH vaccine in different species of animals. We found 
that although GnRH I is highly conserved among species, the immune responses elicited 
by the same vaccine are not quite similar in different species of animals. Moreover, each 
species of animal has its specific reproductive cycles, therefore, accumulating more 
research data on using GnRH-based vaccines for immunocontraception in different species 
of animals is fundamental for optimizing the efficacy and application of such vaccine in 
pets or free-living animals. Otherwise, the side effect on the different animal was also 
different. For example, GonaCon administration in dogs can cause swelling and muscular 
atrophy at the injection site, as well as chronic granulomatous myositis and diffuse 
coagulative necrosis, but few side effects are reported on deer. It is important to evaluate
the vaccine on the target animal.
Point 5: Additionally, previous anti-GnRH vaccines required multiple, which made the 
vaccination cumbersome and difficult to achieve. How do the authors address this concern?Response 5: The GnRH vaccine is a peptide-based subunit vaccine. Compared with other 
modern contraceptive methods, the immunocontraceptive vaccine is safe and easy to 
produce. However, the poor immunogenicity of GnRH peptide has limited its efficacy 
and field application. We understand that a single shot for prolonged infertility is more 
practical for the vaccine to be used in the field. We are currently trying different 
adjuvants, in various combinations to control the release rate. A novel adjuvant that can
have slowly released antigen for prolonging the antibody level is our priority to address 
this issue. The description of limitation was revisied at lines 431-440.
Specific comments:
Point 1: Line 21: Define ICR mice.
Response 1: The ICR mice is the abbreviation of Institute of Cancer Research mice. ICR 
mice are derived from the colony of Swiss mice in the USA. This mouse has been widely 
used in various research fields, including physiology, toxicology, virology, and 
pharmacology. We added the definition of ICR mice in Materials and Methods at line 107. 
Point 2: Line 50: Remove theoretically.
Response 2: Thank you for the suggestion. We revised the sentence and remove the word, 
“theoretically”.
Point 3: Line 55: Rewrite this line “In general, GnRH is highly conserved and typically 
exhibits an identical structure in phylogenetically closed species”
Response 3: Thank you for the comment. We rewrote the sentence to “ High amino acid 
sequence similarity of GnRH is observed in phylogenetically closed species [1].” And 
added the citation to this sentence at lines 72-73.
Point 4: Line 78: Define ISA206.
Response 4: We added the description of adjuvant ISA206. At line 99-100, “Montanide 
ISA206 is a commercial adjuvant based on water-in-oil–water emulsions (W/O/W)” was 
added. 
Point 5: Line 81: sex hormones not sexual hormones
Response 5: Thank you for the kind reminder, we revised the sentence to sex hormone at 
line 102.
Point 6: Line122: no structure is shown in the figure. The line reads obscurely
Response 6: We agreed with your comment. The sentence was changed to “epitope 
arrangement” at line 150. 
Point 7: Line 167: It is not clear what intra and interassays mean.Response 7: Thanks for the reviewer’s suggestion. We added more descriptions of intra 
and inter-assays at lines 200-203. The meaning of these two assays has expressed the 
precision and repeatability of our ELISA experiments. 
The intra-assay described the variation of data from one experiment. The coefficient of 
variation (CV) calculation in intra-assay was the average value from each sample in 
duplicated. The inter-assay was a plate-to-plate consistency. The CV calculation of mean 
values for the low and high concentration standards on each plate. 
Point 8: Line 236: on day 0 not in day 0
Response 8: Thanks for the reviewer’s suggestion. We revised it accordingly at line 279.
Point 9: Line 242: Please explain what intra and interassays are.
Response 9: The calculation of intra and inter-assays have expressed the precision and 
repeatability of our ELISA experiments. According to the acceptable range of reference,
the CV of inter-assay % CVs should less than 15 and intra-assay % CVs should be less 
than 10. In our research, the intra-assay and inter-assay CV% of anti-GnRH IgG and 
hormone concentration measurements were <10% and 10%, respectively. We described it 
more carefully at lines 200-203.
Point 10: What are the effects of this contraceptive vaccine on female mice?
Response 10: According to our previous study, our GnRH vaccine has a significant 
decrease the progesterone and estrogen concentration in female mice compared to the 
control group. However, this manuscript focused on the male mice. Therefore, we did not 
put the result of female mice in the manuscript. 
Point 11: Line 332: “In our study” not “Im our study”
Response 11: Thanks for point out the fault. We revised it accordingly at line 385.
[1] Lee VH, Lee LT, Chow BK. Gonadotropin‐releasing hormone: regulation of the GnRH 
gene. The FEBS journal. 2008;275:5458-78

Please see the attachment for the response.

Sincerely, 

Ai-Mei Chang 

Round 2

Reviewer 1 Report

The authors have made numerous changes in response to reviewer comments.  Some of these changes have introduced many grammatical errors and will require editorial or careful author correction. 

For example, line 25:  change “showed” to “observed”.  The sentence would be improved by re-drafting as follows:  “The vaccine induce a specific immune response by week 5, with peak antibody levels observed by week 13 and declining levels thereafter until the end of the study at Week 24.”

Line 30:  What do the authors mean by “should not be similar”?  Delete “not”?  Change conclusion to read: “Assessment of this GnRH vaccine in different species could assist its development for future applications.”

Many nouns are used in the singular when they should be plural:  For example line 45 - crops; humans.  “ ....population control management strategies are needed for specific species.”

Lines 49-58:  please correct the English expression here. Also note that culling is not always inhumane - it depends on the techniques used.  Culling is mainly considered ineffective because it has only a short-term impact on population densities - there is often rapid recovery of numbers through increased reproduction and immigration. 

Line 57: Immunocontraceptive vaccines as alternatives are not easier - otherwise we would have applied them more successfully than we have to date.  Delete “easier”.

Lines 79-81.  The sentence beginning “And several reproductive inhibitors....” does not make sense. Please revise.  Also, you indicate immune responses may be inconsistent - what do you mean?  It is certainly correct that different vaccines can induce different degrees of immune responses (concentration, and types of antibodies) and longevity.  Side effects are a separate issue which must also be considered for each species.  Please re-draft.

Lines 89-91.  It is incorrect to say that most studies using GnRH vaccines have assessed responses in peripubertal animals.   Sexually mature female mammals have been the main source of animals used in the majority of studies quoted by Massei and Cowan (2014). Please revise.  Perhaps the authors should make it clear what their true goal is  - it seems they are mainly interested in managing the fertility of male Macaca cyclopis with their vaccine.  The authors’ Response 33 in their cover letter provides a better context for their study.  So please use it in your Introduction. It clarifies use of adult male mice and the choice of adjuvant very well.  

Lines 102- onwards.  This is not a strong justification for the use of 30 week-old mice. What is the relevance for 30 week old mice being equivalent to humans of around 30 years?  It is the life cycle of the mouse that is critical, and, in the field, it would be the overall population dynamics that would be the focus of delivery of fertility control.  Hence all life stages could be affected, depending on how the vaccine is delivered.

Line 204:  Change to read as:  After euthanasia, the testes were dissected for morphological measurements.   Do you mean “digital” rather than “electric” caliper?

Line 249:  Delete “determine the”

Please calculate and present testis volumes in your results.  The formula is as follows:

 where;       V = volume (mm3), a = 0.5 x testis width, b = 0.5 x testis length

That is:  Volume (mm3) = 1.33 x 22/7 x (0.5 x testis width)2 x 0.5 x testis length.

This should then be divided by body weight to give a value of mm3/g body weight.  Please change this in the methods (line 253) and in the results (line 283), and in Table 1 (line 319), present only the testis volume/g body weight for treated and control groups.

Line 343:  Change to read as: “The average number of pups per litter was  ...”

Line 352:  References 47-49 are papers published since  2000.  Change this sentence to read as “....as an alternative (see 47-49 for reviews).

Line 358:  reference 20 does not seem the most appropriate one to quote here.

Line 366:  There is a word missing in this sentence  “....a combination of  XXX mixed with...”

Line 369:  deleted second use of “swelling”

Line 381:  modify this sentence by referring to testis volume differences rather than low testis/body weight ratios.

Line 396-397:  Change to read as:  “... also found in mice in a GnRH-based vaccine study.”

Line 402 onwards.  The English expression in this paragraph needs considerable improvement.

Author Response

Dear reviewer 1,

We appreciate your valuable comment. According to reviewers’ comments, we had made extensive modifications to our manuscript and supplemented extra data to make our results convincing.

We explain point-by-point the details of the comments. All modifications in this article had been using the "Track Changes" function in Microsoft Word. We appreciate the time and effort of your help.

Please see the attachment for the revision.

Sincerely,

Ai-Mei Chang
